# QUERY2BOX: REASONING OVER KNOWLEDGE GRAPHS IN VECTOR SPACE USING BOX EMBEDDINGS

**Hongyu Ren**\*, **Weihua Hu**\*, **Jure Leskovec**
Department of Computer Science, Stanford University
{hyren,weihuahu,jure}@cs.stanford.edu

## ABSTRACT

Answering complex logical queries on large-scale incomplete knowledge graphs (KGs) is a fundamental yet challenging task. Recently, a promising approach to this problem has been to embed KG entities as well as the query into a vector space such that entities that answer the query are embedded close to the query. However, prior work models queries as single points in the vector space, which is problematic because a complex query represents a potentially large set of its answer entities, but it is unclear how such a set can be represented as a single point. Furthermore, prior work can only handle queries that use conjunctions ($\wedge$) and existential quantifiers ($\exists$). Handling queries with logical disjunctions ($\vee$) remains an open problem. Here we propose QUERY2BOX, an embedding-based framework for reasoning over arbitrary queries with $\wedge$, $\vee$, and $\exists$ operators in massive and incomplete KGs. Our main insight is that queries can be embedded as boxes (i.e., hyper-rectangles), where a set of points inside the box corresponds to a set of answer entities of the query. We show that conjunctions can be naturally represented as intersections of boxes and also prove a negative result that handling disjunctions would require embedding with dimension proportional to the number of KG entities. However, we show that by transforming queries into a Disjunctive Normal Form, QUERY2BOX is capable of handling arbitrary logical queries with $\wedge$, $\vee$, $\exists$ in a scalable manner. We demonstrate the effectiveness of QUERY2BOX on three large KGs and show that QUERY2BOX achieves up to 25% relative improvement over the state of the art.

## 1 INTRODUCTION

Knowledge graphs (KGs) capture different types of relationships between entities, *e.g.*, *Canada* $\xrightarrow{citizen}$ *Hinton*. Answering arbitrary logical queries, such as "*where did Canadian citizens with Turing Award graduate?*", over such KGs is a fundamental task in question answering, knowledge base reasoning, as well as AI more broadly.

First-order logical queries can be represented as Directed Acyclic Graphs (DAGs) (Fig. 1(A)) and be reasoned according to the DAGs to obtain a set of answers (Fig. 1(C)). While simple and intuitive, such approach has many drawbacks: (1) Computational complexity of subgraph matching is exponential in the query size, and thus cannot scale to modern KGs; (2) Subgraph matching is very sensitive as it cannot correctly answer queries with missing relations. To remedy (2) one could impute missing relations (Koller et al., 2007; Džeroski, 2009; De Raedt, 2008; Nickel et al., 2016) but that would only make the KG denser, which would further exacerbate issue (1) (Dalvi & Suciu, 2007; Krompaß et al., 2014).

Recently, a promising alternative approach has emerged, where logical queries as well as KG entities are embedded into a low-dimensional vector space such that entities that answer the query are embedded close to the query (Guu et al., 2015; Hamilton et al., 2018; Das et al., 2017). Such approach robustly handles missing relations (Hamilton et al., 2018) and is also orders of magnitude faster, as answering an arbitrary logical query is reduced to simply identifying entities nearest to the embedding of the query in the vector space.

---

\*Equal contributions. Project website with data and code: http://snap.stanford.edu/query2box

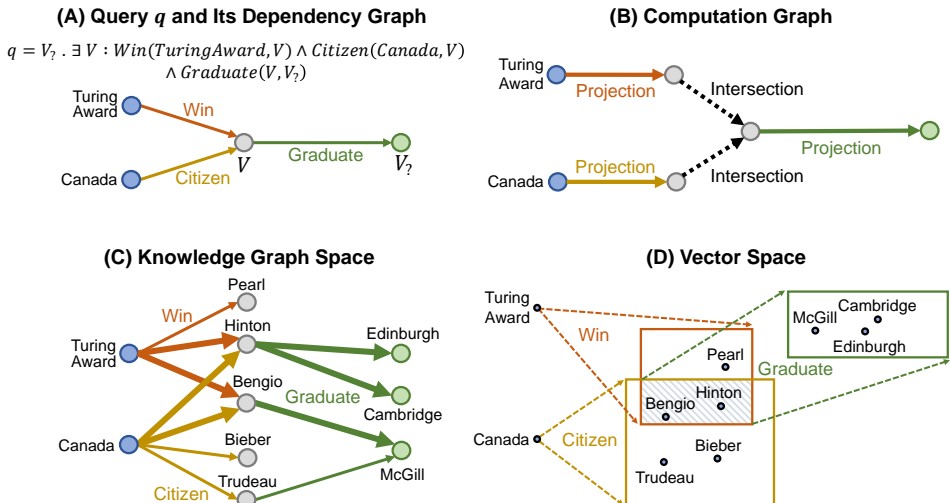

Figure 1: **Query2Box reasoning framework.** **(A)** A given conjunctive query *"Where did Canadian citizens with Turing Award graduate?"* can be represented with a dependency graph. **(B)** Computation graph specifies the reasoning procedure to obtain a set of answers for the query in (A). **(C)** Example knowledge graph, where green nodes/entities denote answers to the query. Bold arrows indicate subgraphs that match the query graph in (A). **(D)** In QUERY2BOX, nodes of the KG are embedded as points in the vector space. We then obtain query embedding according to the computation graph (B) as a sequence of box operations: start with two nodes *TuringAward* and *Canada* and apply *Win* and *Citizen* projection operators, followed by an intersection operator (denoted as a shaded intersection of yellow and orange boxes) and another projection operator. The final embedding of the query is a green box and query's answers are the entities inside the box.

However, prior work embeds a query into a single point in the vector space. This is problematic because answering a logical query requires modeling a set of active entities while traversing the KG (Fig. 1(C)), and how to effectively model a set with a single point is unclear. Furthermore, it is also unnatural to define logical operators (*e.g.*, set intersection) of two points in the vector space. Another fundamental limitation of prior work is that it can only handle *conjunctive queries*, a subset of first-order logic that only involves conjunction ($\land$) and existential quantifier ($\exists$), but not disjunction ($\lor$). It remains an open question how to handle disjunction effectively in the vector space.

Here we present QUERY2BOX, an embedding-based framework for reasoning over KGs that is capable of handling arbitrary Existential Positive First-order (EPFO) logical queries (*i.e.*, queries that include any set of $\land$, $\lor$, and $\exists$) in a scalable manner. First, to accurately model a set of entities, our key idea is to use a closed *region* rather than a single point in the vector space. Specifically, we use a box (axis-aligned hyper-rectangle) to represent a query (Fig. 1(D)). This provides three important benefits: (1) Boxes naturally model sets of entities they enclose; (2) Logical operators (*e.g.*, set intersection) can naturally be defined over boxes similarly as in Venn diagrams (Venn, 1880); (3) Executing logical operators over boxes results in new boxes, which means that the operations are closed; thus, logical reasoning can be efficiently performed in QUERY2BOX by iteratively updating boxes according to the query computation graph (Fig. 1(B)(D)).

We show that QUERY2BOX can naturally handle conjunctive queries. We first prove a negative result that embedding EPFO queries to only single points or boxes is intractable as it would require embedding dimension proportional to the number of KG entities. However, we provide an elegant solution, where we transform a given EPFO logical query into a Disjunctive Normal Form (DNF) (Davey & Priestley, 2002), *i.e.*, disjunction of conjunctive queries. Given any EPFO query, QUERY2BOX represents it as *a set of individual boxes*, where each box is obtained for each conjunctive query in the DNF. We then return nearest neighbor entities to *any of the boxes* as the answers to the query. This means that to answer any EPFO query we first answer individual conjunctive queries and then take the union of the answer entities.

We evaluate QUERY2BOX on three standard KG benchmarks and show: (1) QUERY2BOX provides strong generalization as it can answer complex queries; (2) QUERY2BOX can generalize to new

logical query structures that it has never seen during training; (3) QUERY2BOX is able to implicitly impute missing relations as it can answer any EPFO query with high accuracy even when relations involving answering the query are missing in the KG; (4) QUERY2BOX provides up to 25% relative improvement in accuracy of answering EPFO queries over state-of-the-art baselines.

## 2 FURTHER RELATED WORK

Most related to our work are embedding approaches for multi-hop reasoning over KGs (Bordes et al., 2013; Das et al., 2017; Guu et al., 2015; Hamilton et al., 2018). Crucial difference is that we provide a way to tractably handle a larger subset of the first-order logic (EPFO queries vs. conjunctive queries) and that we embed queries as boxes, which provides better accuracy and generalization.

Second line of related work is on structured embeddings, which associate images, words, sentences, or knowledge base concepts with geometric objects such as regions (Erk, 2009; Vilnis et al., 2018; Li et al., 2019), densities (Vilnis & McCallum, 2014; He et al., 2015; Athiwaratkun & Wilson, 2018), and orderings (Vendrov et al., 2016; Lai & Hockenmaier, 2017; Li et al., 2017). While the above work uses geometric objects to model individual entities and their pairwise relations, we use the geometric objects to model sets of entities and reason over those sets. In this sense our work is also related to classical Venn Diagrams (Venn, 1880), where boxes are essentially the Venn Diagrams in vector space, but our boxes and entity embeddings are jointly learned, which allows us to reason over incomplete KGs.

Box embeddings have also been used to model hierarchical nature of concepts in an ontology with uncertainty (Vilnis et al., 2018; Li et al., 2019). While our work is also based on box embeddings we employ them for logical reasoning in massive heterogeneous knowledge graphs.

## 3 QUERY2BOX: LOGICAL REASONING OVER KGS IN VECTOR SPACE

Here we present the QUERY2BOX, where we will define an objective function that allows us to learn embeddings of entities in the KG, and at the same time also learn parameterized geometric logical operators over boxes. Then given an arbitrary EPFO query $q$ (Fig. 1(A)), we will identify its computation graph (Fig. 1(B)), and embed the query by executing a set of geometric operators over boxes (Fig. 1(D)). Entities that are enclosed in the final box embedding are returned as answers to the query (Fig. 1(D)).

In order to train our system, we generate a set of queries together with their answers at training time and then learn entity embeddings and geometric operators such that queries can be accurately answered. We show in the following sections that our approach is able to generalize to queries and logical structures never seen during training. Furthermore, as we show in experiments, our approach is able to implicitly impute missing relations and answer queries that would be impossible to answer with traditional graph traversal methods.

In the following we first only consider conjunctive queries (conjunction and existential operator) and then we extend our method to also include disjunction.

### 3.1 KNOWLEDGE GRAPHS AND CONJUNCTIVE QUERIES

We denote a KG as $\mathcal{G} = (\mathcal{V}, \mathcal{R})$, where $v \in \mathcal{V}$ represents an entity, and $r \in \mathcal{R}$ is a binary function $r : \mathcal{V} \times \mathcal{V} \to \{\text{True}, \text{False}\}$, indicating whether the relation $r$ holds between a pair of entities or not. In the KG, such binary output indicates the existence of the directed edge between a pair of entities, *i.e.*, $v \xrightarrow{r} v'$ iff $r(v, v') = \text{True}$.

Conjunctive queries are a subclass of the first-order logical queries that use existential ($\exists$) and conjunction ($\wedge$) operations. They are formally defined as follows.

$$q[V_?] = V_? \, . \, \exists V_1, \ldots, V_k : e_1 \wedge e_2 \wedge \ldots \wedge e_n, \tag{1}$$
$$\text{where } e_i = r(v_a, V), V \in \{V_?, V_1, \ldots, V_k\}, v_a \in \mathcal{V}, r \in \mathcal{R},$$
$$\text{or } e_i = r(V, V'), V, V' \in \{V_?, V_1, \ldots, V_k\}, V \neq V', r \in \mathcal{R},$$

where $v_a$ represents non-variable anchor entity, $V_1, \ldots, V_k$ are existentially quantified bound variables, $V_?$ is the target variable. The goal of answering the logical query $q$ is to find a set of entities $[\![q]\!] \subseteq \mathcal{V}$ such that $v \in [\![q]\!]$ iff $q[v] = \text{True}$. We call $[\![q]\!]$ the *denotation set* (*i.e.*, answer set) of query $q$.

As shown in Fig. 1(A), the *dependency graph* is a graphical representation of conjunctive query $q$, where nodes correspond to variable or non-variable entities in $q$ and edges correspond to relations in $q$. In order for the query to be *valid*, the corresponding dependency graph needs to be a Directed Acyclic Graph (DAG), with the anchor entities as the source nodes of the DAG and the query target $V_?$ as the unique sink node (Hamilton et al., 2018).

From the dependency graph of query $q$, one can also derive the *computation graph*, which consists of two types of directed edges that represent operators over sets of entities:

- **Projection:** Given a set of entities $S \subseteq \mathcal{V}$, and relation $r \in \mathcal{R}$, this operator obtains $\cup_{v \in S} A_r(v)$, where $A_r(v) \equiv \{v' \in \mathcal{V} : r(v, v') = \text{True}\}$.
- **Intersection:** Given a set of entity sets $\{S_1, S_2, \ldots, S_n\}$, this operator obtains $\cap_{i=1}^n S_i$.

For a given query $q$, the computation graph specifies the procedure of reasoning to obtain a set of answer entities, *i.e.*, starting from a set of anchor nodes, the above two operators are applied iteratively until the unique sink target node is reached. The entire procedure is analogous to traversing KGs following the computation graph (Guu et al., 2015).

### 3.2 REASONING OVER SETS OF ENTITIES USING BOX EMBEDDINGS

So far we have defined conjunctive queries as computation graphs that can be executed directly over the nodes and edges in the KG. Now, we define logical reasoning in the vector space. Our intuition follows Fig. 1: Given a complex query, we shall decompose it into a sequence of logical operations, and then execute these operations in the vector space. This way we will obtain the embedding of the query, and answers to the query will be entities that are enclosed in the final query embedding box.

In the following, we detail our two methodological advances: (1) the use of box embeddings to efficiently model and reason over sets of entities in the vector space, and (2) how to *tractably* handle disjunction operator ($\vee$), expanding the class of first-order logic that can be modeled in the vector space (Section 3.3).

**Box embeddings.** To efficiently model a set of entities in the vector space, we use *boxes* (*i.e.*, axis-aligned hyper-rectangles). The benefit is that unlike a single point, the box has the *interior*; thus, if an entity is in a set, it is natural to model the entity embedding to be a point *inside* the box. Formally, we operate on $\mathbb{R}^d$, and define a box in $\mathbb{R}^d$ by $\mathbf{p} = (\text{Cen}(\mathbf{p}), \text{Off}(\mathbf{p})) \in \mathbb{R}^{2d}$ as:

$$\text{Box}_{\mathbf{p}} \equiv \{\mathbf{v} \in \mathbb{R}^d : \text{Cen}(\mathbf{p}) - \text{Off}(\mathbf{p}) \preceq \mathbf{v} \preceq \text{Cen}(\mathbf{p}) + \text{Off}(\mathbf{p})\}, \tag{2}$$

where $\preceq$ is element-wise inequality, $\text{Cen}(\mathbf{p}) \in \mathbb{R}^d$ is the center of the box, and $\text{Off}(\mathbf{p}) \in \mathbb{R}^d_{\geq 0}$ is the positive offset of the box, modeling the size of the box. Each entity $v \in \mathcal{V}$ in KG is assigned a single vector $\mathbf{v} \in \mathbb{R}^d$ (*i.e.*, a zero-size box), and the box embedding $\mathbf{p}$ models $\{v \in \mathcal{V} : \mathbf{v} \in \text{Box}_{\mathbf{p}}\}$, *i.e.*, a set of entities whose vectors are inside the box. For the rest of the paper, we use the bold face to denote the embedding, *e.g.*, embedding of $v$ is denoted by $\mathbf{v}$.

Our framework reasons over KGs in the vector space following the computation graph of the query, as shown in Fig. 1(D): we start from the initial box embeddings of the source nodes (anchor entities) and sequentially update the embeddings according to the logical operators. Below, we describe how we set initial box embeddings for the source nodes, as well as how we model projection and intersection operators (defined in Sec. 3.1) as geometric operators that operate over boxes. After that, we describe our entity-to-box distance function and the overall objective that learns embeddings as well as the geometric operators.

**Initial boxes for source nodes.** Each source node represents an anchor entity $v \in \mathcal{V}$, which we can regard as a *set* that only contains the single entity. Such a single-element set can be naturally modeled by a box of size/offset zero centered at $\mathbf{v}$. Formally, we set the initial box embedding as $(\mathbf{v}, \mathbf{0})$, where $\mathbf{v} \in \mathbb{R}^d$ is the anchor entity vector and $\mathbf{0}$ is a $d$-dimensional all-zero vector.

**Geometric projection operator.** We associate each relation $r \in \mathcal{R}$ with relation embedding $\mathbf{r} = (\text{Cen}(\mathbf{r}), \text{Off}(\mathbf{r})) \in \mathbb{R}^{2d}$ with $\text{Off}(\mathbf{r}) \succeq \mathbf{0}$. Given an input box embedding $\mathbf{p}$, we model the projection by $\mathbf{p} + \mathbf{r}$, where we sum the centers and sum the offsets. This gives us a new box with the *translated center* and *larger offset* because $\text{Off}(\mathbf{r}) \succeq \mathbf{0}$, as illustrated in Fig. 2(A). The adaptive box size effectively models a different number of entities/vectors in the set.

**Geometric intersection operator.** We model the intersection of a set of box embeddings $\{\mathbf{p_1}, \ldots, \mathbf{p_n}\}$ as $\mathbf{p}_{\text{inter}} = (\text{Cen}(\mathbf{p}_{\text{inter}}), \text{Off}(\mathbf{p}_{\text{inter}}))$, which is calculated by performing attention

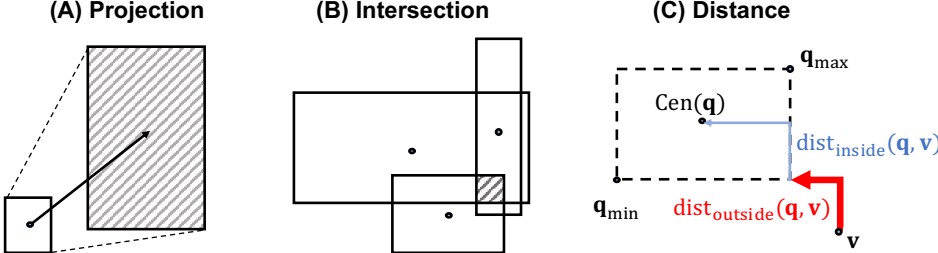

Figure 2: The geometric intuition of the two operations and distance function in QUERY2BOX. **(A)** Projection generates a larger box with a translated center. **(B)** Intersection generates a smaller box lying inside the given set of boxes. **(C)** Distance $\text{dist}_{\text{box}}$ is the weighted sum of $\text{dist}_{\text{outside}}$ and $\text{dist}_{\text{inside}}$, where the latter is weighted less.

over the box centers (Bahdanau et al., 2015) and shrinking the box offset using the sigmoid function:

$$\text{Cen}(\mathbf{p}_{\text{inter}}) = \sum_i \mathbf{a}_i \odot \text{Cen}(\mathbf{p_i}), \quad \mathbf{a}_i = \frac{\exp(\text{MLP}(\mathbf{p_i}))}{\sum_j \exp(\text{MLP}(\mathbf{p_j}))},$$

$$\text{Off}(\mathbf{p}_{\text{inter}}) = \text{Min}(\{\text{Off}(\mathbf{p_1}), \dots, \text{Off}(\mathbf{p_n})\}) \odot \sigma(\text{DeepSets}(\{\mathbf{p_1}, \dots, \mathbf{p_n}\})),$$

where $\odot$ is the dimension-wise product, $\text{MLP}(\cdot) : \mathbb{R}^{2d} \to \mathbb{R}^d$ is the Multi-Layer Perceptron, $\sigma(\cdot)$ is the sigmoid function, $\text{DeepSets}(\cdot)$ is the permutation-invariant deep architecture (Zaheer et al., 2017), and both $\text{Min}(\cdot)$ and $\exp(\cdot)$ are applied in a dimension-wise manner. Following Hamilton et al. (2018), we model all the deep sets by $\text{DeepSets}(\{\mathbf{x_1}, \dots, \mathbf{x_N}\}) = \text{MLP}((1/N) \cdot \sum_{i=1}^N \text{MLP}(\mathbf{x_i}))$, where all the hidden dimensionalities of the two MLPs are the same as the input dimensionality. The intuition behind our geometric intersection is to generate a *smaller* box that lies *inside* a set of boxes, as illustrated in Fig. 2(B).[1] Different from the generic deep sets to model the intersection (Hamilton et al., 2018), our geometric intersection operator effectively constrains the center position and models the shrinking set size.

**Entity-to-box distance.** Given a query box $\mathbf{q} \in \mathbb{R}^{2d}$ and an entity vector $\mathbf{v} \in \mathbb{R}^d$, we define their distance as

$$\text{dist}_{\text{box}}(\mathbf{v}; \mathbf{q}) = \text{dist}_{\text{outside}}(\mathbf{v}; \mathbf{q}) + \alpha \cdot \text{dist}_{\text{inside}}(\mathbf{v}; \mathbf{q}), \tag{3}$$

where $\mathbf{q}_{\text{max}} = \text{Cen}(\mathbf{q}) + \text{Off}(\mathbf{q}) \in \mathbb{R}^d$, $\mathbf{q}_{\text{min}} = \text{Cen}(\mathbf{q}) - \text{Off}(\mathbf{q}) \in \mathbb{R}^d$ and $0 < \alpha < 1$ is a fixed scalar, and

$$\text{dist}_{\text{outside}}(\mathbf{v}; \mathbf{q}) = \|\text{Max}(\mathbf{v} - \mathbf{q}_{\text{max}}, \mathbf{0}) + \text{Max}(\mathbf{q}_{\text{min}} - \mathbf{v}, \mathbf{0})\|_1,$$

$$\text{dist}_{\text{inside}}(\mathbf{v}; \mathbf{q}) = \|\text{Cen}(\mathbf{q}) - \text{Min}(\mathbf{q}_{\text{max}}, \text{Max}(\mathbf{q}_{\text{min}}, \mathbf{v}))\|_1.$$

As illustrated in Fig. 2(C), $\text{dist}_{\text{outside}}$ corresponds to the distance between the entity and closest corner/side of the box. Analogously, $\text{dist}_{\text{inside}}$ corresponds to the distance between the center of the box and its side/corner (or the entity itself if the entity is inside the box).

The key here is to *downweight* the distance inside the box by using $0 < \alpha < 1$. This means that as long as entity vectors are inside the box, we regard them as "close enough" to the query center (*i.e.*, $\text{dist}_{\text{outside}}$ is 0, and $\text{dist}_{\text{inside}}$ is scaled by $\alpha$). When $\alpha = 1$, $\text{dist}_{\text{box}}$ reduces to the ordinary $L_1$ distance, *i.e.*, $\|\text{Cen}(\mathbf{q}) - \mathbf{v}\|_1$, which is used by the conventional TransE (Bordes et al., 2013) as well as prior query embedding methods (Guu et al., 2015; Hamilton et al., 2018).

**Training objective.** Our next goal is to *learn* entity embeddings as well as geometric projection and intersection operators.

Given a training set of queries and their answers, we optimize a negative sampling loss (Mikolov et al., 2013) to effectively optimize our distance-based model (Sun et al., 2019):

$$L = -\log \sigma\left(\gamma - \text{dist}_{\text{box}}(\mathbf{v}; \mathbf{q})\right) - \sum_{i=1}^k \frac{1}{k} \log \sigma\left(\text{dist}_{\text{box}}(\mathbf{v_i'}; \mathbf{q}) - \gamma\right), \tag{4}$$

---

[1]One possible choice here would be to directly use raw box intersection, however, we find that our richer learnable parameterization is more expressive and robust

where $\gamma$ represents a fixed scalar margin, $v \in [\![q]\!]$ is a positive entity (*i.e.*, answer to the query $q$), and $v_i' \notin [\![q]\!]$ is the $i$-th negative entity (non-answer to the query $q$) and $k$ is the number of negative entities.

### 3.3 TRACTABLE HANDLING OF DISJUNCTION USING DISJUNCTIVE NORMAL FORM

So far we have focused on conjunctive queries, and our aim here is to tractably handle in the vector space a wider class of logical queries, called *Existential Positive First-order (EPFO) queries* (Dalvi & Suciu, 2012) that involve $\lor$ in addition to $\exists$ and $\land$. We specifically focus on EPFO queries whose computation graphs are a DAG, same as that of conjunctive queries (Section 3.1), except that we now have an additional type of directed edge, called *union* defined as follows:

- **Union:** Given a set of entity sets $\{S_1, S_2, \ldots, S_n\}$, this operator obtains $\cup_{i=1}^n S_i$.

A straightforward approach here would be to define another geometric operator for union and embed the query as we did in the previous sections. An immediate challenge for our box embeddings is that boxes can be located anywhere in the vector space, so their union would no longer be a simple box. In other words, union operation over boxes is not closed.

Theoretically, we prove a general negative result that holds for *any* embedding-based method that embeds query $q$ into $\mathbf{q}$ and uses some distance function to retrieve entities, *i.e.*, $\mathrm{dist}(\mathbf{v}; \mathbf{q}) \leq \beta$ iff $v \in [\![q]\!]$. Here, $\mathrm{dist}(\mathbf{v}; \mathbf{q})$ is the distance between entity and query embeddings, *e.g.*, $\mathrm{dist}_{\mathrm{box}}(\mathbf{v}; \mathbf{q})$ or $\|\mathbf{v} - \mathbf{q}\|_1$, and $\beta$ is a fixed threshold.

**Theorem 1.** *Consider any $M$ conjunctive queries $q_1, \ldots, q_M$ whose denotation sets $[\![q_1]\!], \ldots, [\![q_M]\!]$ are disjoint with each other, $\forall\, i \neq j$, $[\![q_i]\!] \cap [\![q_j]\!] = \varnothing$. Let $D$ be the VC dimension of the function class $\{\mathrm{sign}(\beta - \mathrm{dist}(\cdot; \mathbf{q})) : \mathbf{q} \in \Xi\}$, where $\Xi$ represents the query embedding space and $\mathrm{sign}(\cdot)$ is the sign function. Then, we need $D \geq M$ to model any EPFO query, i.e., $\mathrm{dist}(\mathbf{v}; \mathbf{q}) \leq \beta \Leftrightarrow v \in [\![q]\!]$ is satisfied for every EPFO query $q$.*

The proof is provided in Appendix A, where the key is that with the introduction of the union operation any subset of denotation sets can be the answer, which forces us to model the *powerset* $\{\cup_{q_i \in S} [\![q_i]\!] : S \subseteq \{q_1, \ldots, q_M\}\}$ in a vector space.

For a real-world KG, there are $M \approx |\mathcal{V}|$ conjunctive queries with non-overlapping answers. For example, in the commonly-used FB15k dataset (Bordes et al., 2013), derived from the Freebase (Bollacker et al., 2008), we find $M = 13{,}365$, while $|\mathcal{V}|$ is 14,951 (see Appendix B for the details).

Theorem 1 shows that in order to accurately model *any* EPFO query with the existing framework, the complexity of the distance function measured by the VC dimension needs to be as large as the number of KG entities. This implies that if we use common distance functions based on hyper-plane, Euclidean sphere, or axis-aligned rectangle,[2] their parameter dimensionality needs to be $\Theta(M)$, which is $\Theta(|\mathcal{V}|)$ for real KGs we are interested in. In other words, the dimensionality of the logical query embeddings needs to be $\Theta(|\mathcal{V}|)$, which is not low-dimensional; thus not scalable to large KGs and not generalizable in the presence of unobserved KG edges.

To rectify this issue, our key idea is to transform a given EPFO query into a Disjunctive Normal Form (DNF) (Davey & Priestley, 2002), *i.e.*, disjunction of conjunctive queries, so that union operation only appears in the last step. Each of the conjunctive queries can then be reasoned in the low-dimensional space, after which we can aggregate the results by a simple and intuitive procedure. In the following, we describe the transformation to DNF and the aggregation procedure.

**Transformation to DNF.** Any first-order logic can be transformed into the equivalent DNF (Davey & Priestley, 2002). We perform such transformation directly in the space of computation graph, *i.e.*, moving all the edges of type "union" to the last step of the computation graph. Let $G_q = (V_q, E_q)$ be the computation graph for a given EPFO query $q$, and let $V_{\mathrm{union}} \subset V_q$ be a set of nodes whose in-coming edges are of type "union". For each $v \in V_{\mathrm{union}}$, define $P_v \subset V_q$ as a set of its parent nodes. We first generate $N = \prod_{v \in V_{\mathrm{union}}} |P_v|$ different computation graphs $G_{q^{(1)}}, \ldots, G_{q^{(N)}}$ as follows, each with different choices of $v_{\mathrm{parent}}$ in the first step.

1. For every $v \in V_{\mathrm{union}}$, select one parent node $v_{\mathrm{parent}} \in P_v$.

---

[2] For the detailed VC dimensions of these function classes, see Vapnik (2013). Crucially, their VC dimensions are all linear with respect to the number of parameters $d$.

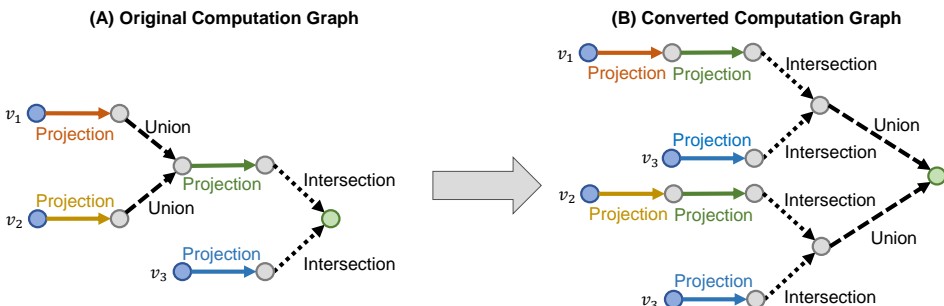

Figure 3: Illustration of converting a computation graph of an EPFO query into an equivalent computation graph of the Disjunctive Normal Form.

    2. Remove all the edges of type 'union.'
    3. Merge $v$ and $v_{\text{parent}}$, while retaining all other edge connections.

We then combine the obtained computation graphs $G_{q^{(1)}}, \ldots, G_{q^{(N)}}$ as follows to give the final equivalent computation graph.

    1. Convert the target sink nodes of all the obtained computation graphs into the existentially quantified bound variables nodes.
    2. Create a new target sink node $V_?$, and draw directed edges of type "union" from all the above variable nodes to the new target node.

An example of the entire transformation procedure is illustrated in Fig. 3. By the definition of the union operation, our procedure gives the equivalent computation graph as the original one. Furthermore, as all the union operators are removed from $G_{q^{(1)}}, \ldots, G_{q^{(N)}}$, all of these computation graphs represent conjunctive queries, which we denote as $q^{(1)}, \ldots, q^{(N)}$. We can then apply existing framework to obtain a set of embeddings for these conjunctive queries as $\mathbf{q^{(1)}}, \ldots, \mathbf{q^{(N)}}$.

**Aggregation.** Next we define the distance function between the given EPFO query $q$ and an entity $v \in \mathcal{V}$. Since $q$ is logically equivalent to $q^{(1)} \vee \cdots \vee q^{(N)}$, we can naturally define the *aggregated* distance function using the box distance $\text{dist}_{\text{box}}$:

$$\text{dist}_{\text{agg}}(\mathbf{v}; q) = \text{Min}(\{\text{dist}_{\text{box}}(\mathbf{v}; \mathbf{q^{(1)}}), \ldots, \text{dist}_{\text{box}}(\mathbf{v}; \mathbf{q^{(N)}})\}), \quad (5)$$

where $\text{dist}_{\text{agg}}$ is parameterized by the EPFO query $q$. When $q$ is a conjunctive query, *i.e.*, $N = 1$, $\text{dist}_{\text{agg}}(\mathbf{v}; q) = \text{dist}_{\text{box}}(\mathbf{v}; \mathbf{q})$. For $N > 1$, $\text{dist}_{\text{agg}}$ takes the minimum distance to the closest box as the distance to an entity. This modeling aligns well with the union operation; an entity is inside the union of sets as long as the entity is in *one of* the sets. Note that our DNF-query rewriting scheme is general and is able to extend any method that works for conjunctive queries (*e.g.*, (Hamilton et al., 2018)) to handle more general class of EPFO queries.

**Computational complexity.** The computational complexity of answering an EPFO query with our framework is equal to that of answering the $N$ conjunctive queries. In practice, $N$ might not be so large, and all the $N$ computations can be parallelized. Furthermore, answering each conjunctive query is very fast as it requires us to execute a sequence of simple box operations (each of which takes constant time) and then perform a range search (Bentley & Friedman, 1979) in the embedding space, which can also be done in constant time using techniques based on Locality Sensitive Hashing (Indyk & Motwani, 1998).

## 4 EXPERIMENTS

Our goal in the experiment section is to evaluate the performance of QUERY2BOX on discovering answers to complex logical queries that cannot be obtained by traversing the incomplete KG. This means, we will focus on answering queries where one or more missing edges in the KG have to be successfully predicted in order to obtain the additional answers.

### 4.1 KNOWLEDGE GRAPHS AND QUERY GENERATION

We perform experiments on three standard KG benchmarks, FB15k (Bordes et al., 2013), FB15k-237 (Toutanova & Chen, 2015), and NELL995 (Xiong et al., 2017) (see Appendix E for NELL995 pre-processing details). Dataset statistics are summarized in Table 5 in Appendix F.

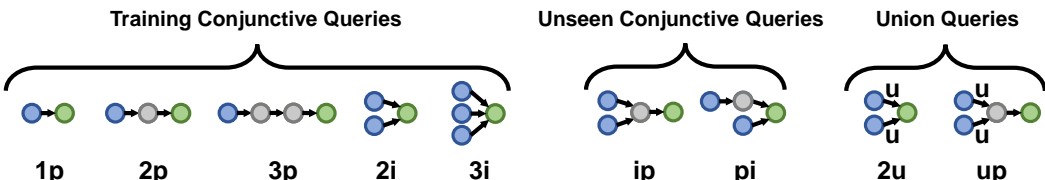

Figure 4: Query structures considered in the experiments, where anchor entities and relations are to be specified to instantiate logical queries. Naming for each query structure is provided under each subfigure, where 'p', 'i', and 'u' stand for 'projection', 'intersection', and 'union', respectively. Models are trained on the first 5 query structures, and evaluated on all 9 query structures. For example, "3p" is a path query of length three, and "2i" is an intersection of cardinality two.

| Dataset | 1p | 2p | 3p | 2i | 3i | ip | pi | 2u | up |
|---------|------|-------|-------|------|------|-------|-------|------|-------|
| FB15k | 10.8 | 255.6 | 250.0 | 90.3 | 64.1 | 593.8 | 190.1 | 27.8 | 227.0 |
| FB15k-237 | 13.3 | 131.4 | 215.3 | 69.0 | 48.9 | 593.8 | 257.7 | 35.6 | 127.7 |
| NELL995 | 8.5 | 56.6 | 65.3 | 30.3 | 15.9 | 310.0 | 144.9 | 14.4 | 62.5 |

Table 1: Average number of answer entities of test queries with missing edges grouped by different query structures (for a KG with 10% edges missing).

We follow the standard evaluation protocol in KG literture: Given the standard split of edges into training, test, and validation sets, we first augment the KG to also include inverse relations and effectively double the number of edges in the graph. We then create three graphs: $\mathcal{G}_{\text{train}}$, which only contains *training* edges and we use this graph to train node embeddings as well as box operators. We then also generate two bigger graphs: $\mathcal{G}_{\text{valid}}$, which contains $\mathcal{G}_{\text{train}}$ plus the validation edges, and $\mathcal{G}_{\text{test}}$, which includes $\mathcal{G}_{\text{valid}}$ as well as the test edges.

We consider 9 kinds of diverse query structures shown and named in Fig. 4. We use 5 query structures for training and then evaluate on all the 9 query structures. We refer the reader to Appendix D for full details on query generation and Table 6 in Appendix F for statistics of the generated logical queries. Given a query $q$, let $[\![q]\!]_{\text{train}}$, $[\![q]\!]_{\text{val}}$, and $[\![q]\!]_{\text{test}}$ denote a set of answer entities obtained by running subgraph matching of $q$ on $\mathcal{G}_{\text{train}}$, $\mathcal{G}_{\text{valid}}$, and $\mathcal{G}_{\text{test}}$, respectively. At the training time, we use $[\![q]\!]_{\text{train}}$ as positive examples for the query and other random entities as negative examples. However, at the test/validation time we proceed differently. Note that we focus on answering queries where generalization performance is crucial and at least one edge needs to be imputed in order to answer the queries. Thus, rather than evaluating a given query on the full validation (or test) set $[\![q]\!]_{\text{val}}$ ($[\![q]\!]_{\text{test}}$) of answers, we validate the method only on *answers that include missing relations*. Given how we constructed $\mathcal{G}_{\text{train}} \subseteq \mathcal{G}_{\text{valid}} \subseteq \mathcal{G}_{\text{test}}$, we have $[\![q]\!]_{\text{train}} \subseteq [\![q]\!]_{\text{val}} \subseteq [\![q]\!]_{\text{test}}$ and thus we evaluate the method on $[\![q]\!]_{\text{val}} \backslash [\![q]\!]_{\text{train}}$ to tune hyper-parameters and then report results identifying answer entities in $[\![q]\!]_{\text{test}} \backslash [\![q]\!]_{\text{val}}$. This means we always evaluate on queries/entities that were not part of the training set and the method has not seen them before. Furthermore, for these queries, traditional graph traversal techniques would not be able to find the answers (due to missing relations).

Table 1 shows the average number of answer entities for different query structures. We observe that complex logical queries (especially 2p, 3p, ip, pi, up) indeed require modeling a much larger number of answer entities (often more than 10 times) than the simple 1p queries do. Therefore, we expect our box embeddings to work particularly well in handling complex queries with many answer entities.[3]

### 4.2 EVALUATION PROTOCOL

Given a test query $q$, for each of its non-trivial answers $v \in [\![q]\!]_{\text{test}} \backslash [\![q]\!]_{\text{val}}$, we use $\text{dist}_{\text{box}}$ in Eq. 3 to rank $v$ among $\mathcal{V} \backslash [\![q]\!]_{\text{test}}$. Denoting the rank of $v$ by $\text{Rank}(v)$, we then calculate evaluation metrics for answering query $q$, such as Mean Reciprocal Rank (MRR) and Hits at $K$ (H@$K$):

$$\text{Metrics}(q) = \frac{1}{|[\![q]\!]_{\text{test}} \backslash [\![q]\!]_{\text{val}}|} \sum_{v \in [\![q]\!]_{\text{test}} \backslash [\![q]\!]_{\text{val}}} f_{\text{metrics}}(\text{Rank}(v)), \tag{6}$$

---

[3]On the simple link prediction (1p query) task, box embeddings provide minor empirical performance improvement over TransE, possibly because simple link prediction does *not* require modeling large sets of entities, as shown in Table 1. See Appendix C for full experimental results on link prediction.

| Method | Avg | 1p | 2p | 3p | 2i | 3i | ip | pi | 2u | up |
|--------|-----|-----|-----|-----|-----|-----|-----|-----|-----|-----|
| **FB15k** | | | | | | | | | | |
| Q2B | **0.484** | **0.786** | **0.413** | **0.303** | **0.593** | **0.712** | **0.211** | **0.397** | **0.608** | **0.33** |
| GQE | 0.386 | 0.636 | 0.345 | 0.248 | 0.515 | 0.624 | 0.151 | 0.310 | 0.376 | 0.273 |
| GQE-DOUBLE | 0.384 | 0.630 | 0.346 | 0.250 | 0.515 | 0.611 | 0.153 | 0.320 | 0.362 | 0.271 |
| **FB15k-237** | | | | | | | | | | |
| Q2B | **0.268** | **0.467** | **0.24** | **0.186** | **0.324** | **0.453** | **0.108** | **0.205** | **0.239** | **0.193** |
| GQE | 0.228 | 0.402 | 0.213 | 0.155 | 0.292 | 0.406 | 0.083 | 0.17 | 0.169 | 0.163 |
| GQE-DOUBLE | 0.23 | 0.405 | 0.213 | 0.153 | 0.298 | 0.411 | 0.085 | 0.182 | 0.167 | 0.16 |
| **NELL995** | | | | | | | | | | |
| Q2B | **0.306** | **0.555** | **0.266** | **0.233** | **0.343** | **0.48** | **0.132** | **0.212** | **0.369** | **0.163** |
| GQE | 0.247 | 0.418 | 0.228 | 0.205 | 0.316 | 0.447 | 0.081 | 0.186 | 0.199 | 0.139 |
| GQE-DOUBLE | 0.248 | 0.417 | 0.231 | 0.203 | 0.318 | 0.454 | 0.081 | 0.188 | 0.2 | 0.139 |

Table 2: H@3 results of QUERY2BOX vs. GQE on FB15k, FB15k-237 and NELL995.

where $f_{\text{metrics}}(x) = \frac{1}{x}$ for MRR, and $f_{\text{metrics}}(x) = \mathbf{1}[x \leq K]$ for H@$K$.

We then average Eq. 6 over all the queries within the same query structure,[4] and report the results separately for different query structures. The same evaluation protocol is applied to the validation stage except that we evaluate on $[\![q]\!]_{\text{val}} \backslash [\![q]\!]_{\text{train}}$ rather than $[\![q]\!]_{\text{test}} \backslash [\![q]\!]_{\text{val}}$.

## 4.3 BASELINE AND MODEL VARIANTS

We compare our framework QUERY2BOX against the state-of-the-art GQE (Hamilton et al., 2018). GQE embeds a query to a single vector, and models projection and intersection operators as translation and deep sets (Zaheer et al., 2017), respectively. The $L_1$ distance is used as the distance between query and entity vectors. For a fair comparison, we also compare with GQE-DOUBLE (GQE with doubled embedding dimensionality) so that QUERY2BOX and GQE-DOUBLE have the same amount of parameters. Refer to Appendix G for the model hyper-parameters used in our experiments. Although the original GQE cannot handle EPFO queries, we apply our DNF-query rewriting strategy and in our evaluation extend GQE to handle general EPFO queries as well. Furthermore, we perform extensive ablation study by considering several variants of QUERY2BOX (abbreviated as Q2B). We list our method as well as its variants below.

- Q2B (**our method**): The box embeddings are used to model queries, and the attention mechanism is used for the intersection operator.
- Q2B-AVG: The attention mechanism for intersection is replaced with averaging.
- Q2B-DEEPSETS: The attention mechanism for intersection is replaced with the deep sets.
- Q2B-AVG-1P: The variant of Q2B-AVG that is trained with only 1p queries (see Fig. 4); thus, logical operators are *not* explicitly trained.
- Q2B-SHAREDOFFSET; The box offset is shared across all queries (every query is represented by a box with the same trainable size).

## 4.4 MAIN RESULTS

We start by comparing our Q2B with state-of-the-art query embedding method GQE (Hamilton et al., 2018) on FB15k, FB15k-237, and NELL995. As listed in Tables 2, our method significantly and consistently outperforms the state-of-the-art baseline across all the query structures, including those not seen during training as well as those with union operations. On average, we obtain 9.8% (25% relative), 3.8% (15% relative), and 5.9% (24% relative) higher H@3 than the best baselines on FB15k, FB15k-237, and NELL995, respectively. Notice that naïvely increasing embedding dimensionality in GQE yields limited performance improvement. Our Q2B is able to effectively model a large set of entities by using the box embedding, and achieves a significant performance gain compared with GQE-DOUBLE (with same number of parameters) that represents queries as point vectors. Also notice

---

[4]Note that our evaluation metric is slightly different from conventional metric (Nickel et al., 2016; Hamilton et al., 2018; Guu et al., 2015), where average is taken over query-answer pairs. The conventional metric is problematic as it can be significantly biased toward correctly answering generic queries with huge number of answers, while dismissing fine-grained queries with a few answers. Here, to treat queries equally regardless of the number of answers they have, we take average *over queries*.

| Method | Avg | 1p | 2p | 3p | 2i | 3i | ip | pi | 2u | up |
|--------|-----|----|----|----|----|----|----|----|----|----|
| **FB15k** | | | | | | | | | | |
| Q2B | **0.484** | 0.786 | **0.413** | **0.303** | **0.593** | **0.712** | **0.211** | **0.397** | 0.608 | **0.330** |
| Q2B-AVG | 0.468 | 0.779 | 0.407 | 0.300 | 0.577 | 0.673 | 0.199 | 0.345 | 0.607 | 0.326 |
| Q2B-DEEPSETS | 0.467 | 0.755 | 0.407 | 0.294 | 0.588 | 0.699 | 0.197 | 0.378 | 0.562 | 0.324 |
| Q2B-AVG-1P | 0.385 | **0.812** | 0.262 | 0.173 | 0.463 | 0.529 | 0.126 | 0.263 | **0.653** | 0.187 |
| Q2B-SHAREDOFFSET | 0.372 | 0.684 | 0.335 | 0.232 | 0.442 | 0.559 | 0.144 | 0.282 | 0.417 | 0.252 |
| **FB15k-237** | | | | | | | | | | |
| Q2B | **0.268** | **0.467** | 0.24 | **0.186** | **0.324** | **0.453** | **0.108** | **0.205** | 0.239 | **0.193** |
| Q2B-AVG | 0.249 | 0.462 | 0.242 | 0.182 | 0.278 | 0.391 | 0.101 | 0.158 | 0.236 | 0.189 |
| Q2B-DEEPSETS | 0.259 | 0.458 | **0.243** | **0.186** | 0.303 | 0.432 | 0.104 | 0.187 | 0.231 | 0.190 |
| Q2B-AVG-1P | 0.219 | 0.457 | 0.193 | 0.132 | 0.251 | 0.319 | 0.083 | 0.142 | **0.241** | 0.152 |
| Q2B-SHAREDOFFSET | 0.207 | 0.391 | 0.199 | 0.139 | 0.251 | 0.354 | 0.082 | 0.154 | 0.15 | 0.142 |
| **NELL995** | | | | | | | | | | |
| Q2B | **0.306** | 0.555 | **0.266** | **0.233** | **0.343** | **0.480** | **0.132** | **0.212** | 0.369 | **0.163** |
| Q2B-AVG | 0.283 | 0.543 | 0.250 | 0.228 | 0.300 | 0.403 | 0.116 | 0.188 | 0.36 | 0.161 |
| Q2B-DEEPSETS | 0.293 | 0.539 | 0.26 | 0.231 | 0.317 | 0.467 | 0.11 | 0.202 | 0.349 | 0.16 |
| Q2B-AVG-1P | 0.274 | **0.607** | 0.229 | 0.182 | 0.277 | 0.315 | 0.097 | 0.18 | **0.443** | 0.133 |
| Q2B-SHAREDOFFSET | 0.237 | 0.436 | 0.219 | 0.201 | 0.278 | 0.379 | 0.096 | 0.174 | 0.217 | 0.137 |

Table 3: H@3 results of QUERY2BOX vs. several variants on FB15k, FB15k-237 and NELL995.

that Q2B performs well on new queries with the same structure as the training queries as well as on new query structures never seen during training, which demonstrates that Q2B generalizes well within and beyond query structures.

We also conduct extensive ablation studies (Tables 3). We summarize the results as follows:

**Importance of attention mechanism.** First, we show that our modeling of intersection using the attention mechanism is important. Given a set of box embeddings $\{\mathbf{p_1}, \ldots, \mathbf{p_n}\}$, Q2B-AVG is the most naïve way to calculate the center of the resulting box embedding $\mathbf{p}_{\text{inter}}$ while Q2B-DEEPSETS is too flexible and neglects the fact that the center should be a weighted average of $\text{Cen}(\mathbf{p_1}), \ldots, \text{Cen}(\mathbf{p_n})$. Compared with the two methods, Q2B achieves better performance in answering queries that involve intersection operation, *e.g.*, 2i, 3i, pi, ip. Specifically, on FB15k-237, Q2B obtains more than 4% and 2% absolute gain in H@3 compared to Q2B-AVG and Q2B-DEEPSETS, respectively.

**Necessity of training on complex queries.** Second, we observe that explicitly training on complex logical queries beyond one-hop path queries (1p in Fig. 4) improves the reasoning performance. Although Q2B-AVG-1P is able to achieve strong performance on 1p and 2u, where answering 2u is essentially answering two 1p queries with an additional minimum operation (see Eq. 5 in Section 3.3), Q2B-AVG-1P fails miserably in answering other types of queries involving logical operators. On the other hand, other methods (Q2B, Q2B-AVG, and Q2B-DEEPSETS) that are explicitly trained on the logical queries achieve much higher accuracy, with up to 10% absolute average improvement of H@3 on FB15k.

**Adaptive box size for different queries.** Third, we investigate the importance of learning *adaptive* offsets (box size) for different queries. Q2B-SHAREDOFFSET is a variant of our Q2B where all the box embeddings *share* the same learnable offset. Q2B-SHAREDOFFSET does not work well on all types of queries. This is most likely because different queries have different numbers of answer entities, and the adaptive box size enables us to better model it. In fact, we find that box offset varies significantly across different relations, and one-to-many relations tend to have larger offset embeddings (see Appendix H for the details).

## 5 CONCLUSION

In this paper we proposed a reasoning framework called QUERY2BOX that can effectively model and reason over sets of entities as well as handle EPFO queries in a vector space. Given a logical query, we first transform it into DNF, embed each conjunctive query into a box, and output entities closest to their nearest boxes. Our approach is capable of handling all types of EPFO queries scalably and accurately. Experimental results on standard KGs demonstrate that QUERY2BOX significantly outperforms the existing work in answering diverse logical queries.

ACKNOWLEDGMENTS

We thank William Hamilton, Rex Ying, and Jiaxuan You for their helpful discussion. W.H is supported by Funai Overseas Scholarship and Masason Foundation Fellowship. J.L is a Chan Zuckerberg Biohub investigator. We gratefully acknowledge the support of DARPA under Nos. FA865018C7880 (ASED), N660011924033 (MCS); ARO under Nos. W911NF-16-1-0342 (MURI), W911NF-16-1-0171 (DURIP); NSF under Nos. OAC-1835598 (CINES), OAC-1934578 (HDR); Stanford Data Science Initiative, Wu Tsai Neurosciences Institute, Chan Zuckerberg Biohub, JD.com, Amazon, Boeing, Docomo, Huawei, Hitachi, Observe, Siemens, UST Global.

The U.S. Government is authorized to reproduce and distribute reprints for Governmental purposes notwithstanding any copyright notation thereon. Any opinions, findings, and conclusions or recommendations expressed in this material are those of the authors and do not necessarily reflect the views, policies, or endorsements, either expressed or implied, of DARPA, NIH, ARO, or the U.S. Government.

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

## A  PROOF OF THEOREM 1

*Proof.* To model *any* EPFO query, we need to at least model a subset of EPFO queries $\mathcal{Q} = \{\vee_{q_i \in S} q_i : S \subseteq \{q_1, \ldots, q_M\}\}$, where the corresponding denotation sets are $\{\cup_{q_i \in S} [\![q_i]\!] : S \subseteq \{q_1, \ldots, q_M\}\}$. For the sake of modeling $\mathcal{Q}$, without loss of generality, we consider assigning a single entity embedding $\mathbf{v_{q_i}}$ to all $v \in [\![q_i]\!]$, so there are $M$ kinds of entity vectors, $\mathbf{v_{q_1}}, \ldots, \mathbf{v_{q_M}}$. To model all queries in $\mathcal{Q}$, it is necessary to satisfy the following.

$$\exists \mathbf{v_{q_1}}, \ldots, \exists \mathbf{v_{q_M}}, \forall S \subseteq \{q_1, \ldots, q_M\}, \exists \mathbf{q_S} \in \Xi, \text{ such that } \text{dist}(\mathbf{v_{q_i}}; \mathbf{q_S}) \begin{cases} \leq \beta & \text{if } q_i \in S, \\ > \beta & \text{if } q_i \notin S. \end{cases} \quad (7)$$

where $\mathbf{q_S}$ is the embedding of query $\vee_{q_i \in S} q_i$. Eq. 7 means that we can learn the $M$ kinds of entity vectors such that for every query in $\mathcal{Q}$, we can obtain its embedding to model the corresponding set using the distance function. Notice that this is agnostic to the specific algorithm to embed query $\vee_{q \in S} q$ into $\mathbf{q_S}$; thus, our result is generally applicable to *any* method that embeds the query into a single vector.

Crucially, satisfying Eq. 7 is equivalent to $\{\text{sign}(\beta - \text{dist}(\cdot; \mathbf{q})) : \mathbf{q} \in \Xi\}$ being able to shutter $\{\mathbf{v_{q_1}}, \ldots, \mathbf{v_{q_M}}\}$, *i.e.*, any binary labeling of the points can be perfectly fit by some classifier in the function class. To sum up, in order to model any EPFO query, we need to at least model any query in $\mathcal{Q}$, which requires the VC dimension of the distance function to be larger than or equal to $M$. □

## B  DETAILS ABOUT COMPUTING $M$ IN THEOREM 1

Given the full KG $\mathcal{G}_{\text{test}}$ for the FB15k dataset, our goal is to find conjunctive queries $q_1, \ldots, q_M$ such that $[\![q_1]\!], \ldots, [\![q_M]\!]$ are disjoint with each other. For conjunctive queries, we use two types of queries: '1p' and '2i' whose query structures are shown in Figure 4. On the FB15k, we instantiate 308,006 queries of type '1p', which we denote by $S_{1\text{p}}$. Out of all the queries in $S_{1\text{p}}$, 129,717 queries have more than one answer entities, and we denote such a set of the queries by $S'_{1\text{p}}$. We then generate a set of queries of type '2i' by first randomly sampling two queries from $S'_{1\text{p}}$ and then taking conjunction; we denote the resulting set of queries by $S_{2\text{i}}$.

Now, we use $S_{1\text{p}}$ and $S_{2\text{i}}$ to generate a set of conjunctive queries whose denotation sets are disjoint with each other. First, we prepare two empty sets $\mathcal{V}_{\text{seen}} = \varnothing$, and $\mathcal{Q} = \varnothing$. Then, for every $q \in S_{1\text{p}}$, if $\mathcal{V}_{\text{seen}} \cap [\![q]\!] = \varnothing$ holds, we let $\mathcal{Q} \leftarrow \mathcal{Q} \cup \{q\}$ and $\mathcal{V}_{\text{seen}} \leftarrow \mathcal{V}_{\text{seen}} \cup [\![q]\!]$. This procedure already gives us $\mathcal{Q}$, where we have $10,812$ conjunctive queries whose denotation sets are disjoint with each other. We can further apply the analogous procedure for $S_{2\text{i}}$, which gives us a further increased $\mathcal{Q}$, where we have $13,365$ conjunctive queries whose denotation sets are disjoint with each other. Therefore, we get $M = 13,365$.

## C  EXPERIMENTS ON LINK PREDICTION

| | FB15k | | FB15k-237 | | NELL995 | |
|---|---|---|---|---|---|---|
| Method | H@3 | MRR | H@3 | MRR | H@3 | MRR |
| query2box | 0.613 | 0.516 | 0.331 | 0.295 | 0.382 | 0.303 |
| query2box-1p | 0.633 | 0.531 | 0.323 | 0.292 | 0.415 | 0.320 |
| TransE | 0.611 | 0.522 | 0.318 | 0.289 | 0.413 | 0.320 |

Table 4: Performance comparison on the simple link prediction task on the three datasets.

In Table 4, we report the link prediction performance (no multi-hop logical reasoning required) following the conventional metrics (taking average over the triples of head, relation, and tail). Here query2box is trained on all five query structures as shown in Figure 4, and query2box-1p is only trained on simple 1p queries. We found that our query2box is comparable or slightly better than TransE on simple link prediction. Note that in the case of simple link prediction, we do not expect a huge performance gain by using box embeddings as link prediction does not involve logical reasoning nor handling a large set of answer entities. Also, we see that even if we train query2box over diverse queries, its performance on link prediction is still comparable to TransE and query2box-1p, which are trained solely on the link prediction task.

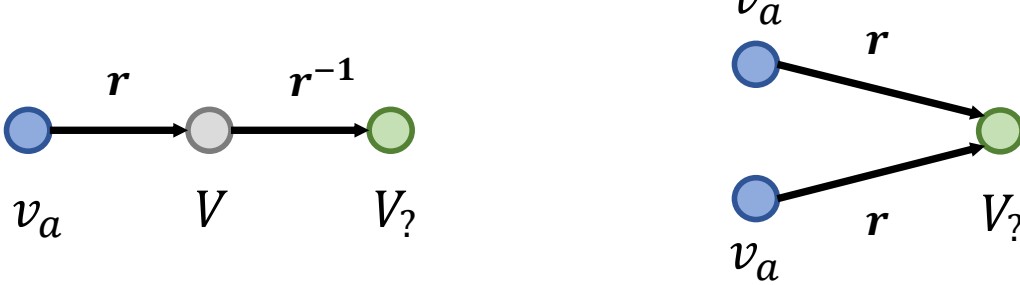

Figure 5: Example of the degenerated queries, including (1) $r$ and $r^{-1}$ appear along one path and (2) same anchor node and relation in intersections.

## D    DETAILS ON QUERY GENERATION

Given $\mathcal{G}_{\text{train}}$, $\mathcal{G}_{\text{valid}}$, and $\mathcal{G}_{\text{test}}$ as defined in Section 4.1, we generate training, validation and test queries of different query structures. During training, we consider the first 5 kinds of query structures. For evaluation, we consider all the 9 query structures in Fig. 4, containing query structures that are both seen and unseen during training time. We instantiate queries in the following way.

Given a KG and a query structure (which is a DAG), we use pre-order traversal to assign an entity and a relation to each node and edge in the DAG of query structure to instantiate a query. Namely, we start from the root of the DAG (which is the target node), we sample an entity $e$ uniformly from the KG to be the root, then for every node connected to the root in the DAG, we choose a relation $r$ uniformly from the in-coming relations of $e$ in the KG, and a new entity $e'$ from the set of entities that reaches $e$ by $r$ in the KG. Then we assign the relation $r$ to the edge and $e'$ to the node, and move on the process based on the pre-order traversal. This iterative process stops after we assign an entity and relation to every node and edge in DAG. The leaf nodes in the DAG serve as the anchor nodes. Note that during the entity and relation assignment, we specifically filter out all the degenerated queries, as shown in Fig. D. Then we perform a post-order traversal of the DAG on the KG, starting from the anchor nodes, to obtain a set of answer entities to this query.

When generating validation/test queries, we explicitly filter out trivial queries that can be fully answered by subgraph matching on $\mathcal{G}_{\text{train}}$/$\mathcal{G}_{\text{valid}}$.

## E    DETAILS OF NELL995 DATASET

Here we detail our pre-processing of the NELL995 dataset, which is originally presented by Xiong et al. (2017). Following Allen et al. (2019), we first combine the validation and test sets with the training set to create the whole knowledge graph for NELL995. Then we create new validation and test set splits by randomly selecting 20,000 triples each from the whole knowledge graph. Note that we filter out all the entities that only appear in the validation and test sets but not in the training set.

| Dataset | Entities | Relations | Training Edges | Validation Edges | Test Edges | Total Edges |
|---------|----------|-----------|----------------|------------------|------------|-------------|
| FB15k | 14,951 | 1,345 | 483,142 | 50,000 | 59,071 | 592,213 |
| FB15k-237 | 14,505 | 237 | 272,115 | 17,526 | 20,438 | 310,079 |
| NELL995 | 63,361 | 200 | 114,213 | 14,324 | 14,267 | 142,804 |

Table 5: Knowledge graph dataset statistics as well as the split into training, validation, and test sets.

| Queries | Training | | Validation | | Test | |
|---------|----------|--------|------------|--------|------|--------|
| Dataset | 1p | others | 1p | others | 1p | others |
| FB15k | 273,710 | 273,710 | 59,097 | 8,000 | 67,016 | 8,000 |
| FB15k-237 | 149,689 | 149,689 | 20,101 | 5,000 | 22,812 | 5,000 |
| NELL995 | 107,982 | 107,982 | 16,927 | 4,000 | 17,034 | 4,000 |

Table 6: Number of training, validation, and test queries generated for different query structures.

## F    DATASET STATISTICS

Table 5 summarizes the basic statistics of the three datasets used in our experiments. Table 6 summarizes the basic statistics of the generated logical queries.

## G    HYPER-PARAMETERS

We use embedding dimensionality of $d = 400$ and set $\gamma = 24$, $\alpha = 0.2$ for the loss in Eq. 4. We train all types of training queries jointly. In every iteration, we sample a minibatch size of 512 queries for each query structure (details in Appendix D), and we sample 1 answer entity and 128 negative entities for each query. We optimize the loss in Eq. 4 using Adam Optimizer (Kingma & Ba, 2015) with learning rate = 0.0001. We train all models for 250 epochs, monitor the performance on the validation set, and report the test performance.

## H    ANALYSIS OF LEARNED BOX OFFSET SIZE

Here we study the correlation between the box size (measured by the L1 norm of the box offset) and the average number of entities that are contained in 1p queries using the corresponding relation. Table 7 shows the top 10 relations with smallest/largest box sizes. We observe a clear trend that the size of the box has a strong correlation with the number of entities the box encloses. Specifically, we see that one-to-many relations tend to have larger offset embeddings, which demonstrates that larger boxes are indeed used to model sets of more points (entities).

| Top 10 relations with smallest box size | #Ent | Box size | Top 10 relations with largest box size | #Ent | Box size |
|-----------------------------------------|------|----------|----------------------------------------|------|----------|
| /architecture/.../owner | 1.0 | 2.3 | /common/.../topic | 3616.0 | 147.0 |
| /base/.../dog_breeds | 2.0 | 4.0 | /user/...taxonomy | 1.0 | 137.2 |
| /education/.../campuses | 1.0 | 4.3 | /common/.../category | 1.3 | 125.6 |
| /education/.../educational_institution | 1.0 | 4.6 | /base/.../administrative_area_type | 1.0 | 123.6 |
| /base/.../collective | 1.0 | 5.1 | /medicine/.../legal_status | 1.5 | 114.9 |
| /base/.../member | 1.0 | 5.1 | /people/.../spouse | 889.8 | 114.3 |
| /people/.../appointed_by | 1.0 | 5.2 | /sports/.../team | 397.9 | 113.9 |
| /base/.../fashion_models_with_this_hair_color | 2.0 | 5.2 | /people/.../location_of_ceremony | 132.0 | 108.4 |
| /fictional_universe/.../parents | 1.0 | 5.5 | /sports/.../team | 83.1 | 104.5 |
| /american_football/.../team | 2.0 | 6.7 | /user/.../subject | 495.0 | 104.2 |

Table 7: Top 10 relations with smallest/largest box size in FB15k.

# I  MRR RESULTS

| Method | Avg | 1p | 2p | 3p | 2i | 3i | ip | pi | 2u | up |
|---|---|---|---|---|---|---|---|---|---|---|
| **FB15k** | | | | | | | | | | |
| Q2B | **0.41** | **0.654** | **0.373** | **0.274** | **0.488** | **0.602** | **0.194** | **0.339** | **0.468** | **0.301** |
| GQE | 0.328 | 0.505 | 0.320 | 0.218 | 0.439 | 0.536 | 0.139 | 0.272 | 0.3 | 0.244 |
| GQE-DOUBLE | 0.326 | 0.49 | 0.3 | 0.222 | 0.438 | 0.532 | 0.142 | 0.28 | 0.285 | 0.242 |
| **FB15k-237** | | | | | | | | | | |
| Q2B | **0.235** | **0.4** | **0.225** | **0.173** | **0.275** | **0.378** | **0.105** | **0.18** | **0.198** | **0.178** |
| GQE | 0.203 | 0.346 | 0.193 | 0.145 | 0.25 | 0.355 | 0.086 | 0.156 | 0.145 | 0.151 |
| GQE-DOUBLE | 0.205 | 0.346 | 0.191 | 0.144 | 0.258 | 0.361 | 0.087 | 0.164 | 0.144 | 0.149 |
| **NELL995** | | | | | | | | | | |
| Q2B | **0.254** | **0.413** | **0.227** | **0.208** | **0.288** | **0.414** | **0.125** | **0.193** | **0.266** | **0.155** |
| GQE | 0.21 | 0.311 | 0.193 | 0.175 | 0.273 | 0.399 | 0.078 | 0.168 | 0.159 | 0.13 |
| GQE-DOUBLE | 0.211 | 0.309 | 0.192 | 0.174 | 0.275 | 0.408 | 0.08 | 0.17 | 0.156 | 0.129 |

Table 8: MRR results of QUERY2BOX vs. GQE on FB15k, FB15k-237 and NELL995.

| Method | Avg | 1p | 2p | 3p | 2i | 3i | ip | pi | 2u | up |
|---|---|---|---|---|---|---|---|---|---|---|
| **FB15k** | | | | | | | | | | |
| Q2B | **0.41** | 0.654 | **0.373** | **0.274** | 0.488 | 0.602 | **0.194** | **0.339** | 0.468 | **0.301** |
| Q2B-AVG | 0.396 | 0.648 | 0.368 | 0.27 | 0.476 | 0.564 | 0.182 | 0.295 | 0.465 | 0.3 |
| Q2B-DEEPSETS | 0.402 | 0.631 | 0.371 | 0.269 | **0.499** | **0.605** | 0.181 | 0.325 | 0.437 | 0.298 |
| Q2B-AVG-1P | 0.324 | **0.688** | 0.236 | 0.159 | 0.378 | 0.435 | 0.122 | 0.225 | **0.498** | 0.178 |
| Q2B-SHAREDOFFSET | 0.296 | 0.511 | 0.273 | 0.199 | 0.351 | 0.444 | 0.132 | 0.233 | 0.311 | 0.213 |
| **FB15k-237** | | | | | | | | | | |
| Q2B | **0.235** | 0.4 | **0.225** | **0.173** | **0.275** | **0.378** | **0.105** | **0.18** | 0.198 | **0.178** |
| Q2B-AVG | 0.219 | 0.398 | 0.222 | 0.171 | 0.236 | 0.328 | 0.1 | 0.145 | 0.193 | 0.177 |
| Q2B-DEEPSETS | 0.23 | 0.395 | 0.224 | 0.172 | 0.264 | 0.372 | 0.101 | 0.168 | 0.194 | 0.176 |
| Q2B-AVG-1P | 0.196 | **0.41** | 0.18 | 0.122 | 0.217 | 0.274 | 0.085 | 0.127 | **0.209** | 0.145 |
| Q2B-SHAREDOFFSET | 0.18 | 0.328 | 0.18 | 0.131 | 0.207 | 0.289 | 0.083 | 0.136 | 0.135 | 0.132 |
| **NELL995** | | | | | | | | | | |
| Q2B | **0.254** | 0.413 | **0.227** | **0.208** | **0.288** | **0.414** | **0.125** | **0.193** | 0.266 | **0.155** |
| Q2B-AVG | 0.235 | 0.406 | 0.219 | 0.2 | 0.251 | 0.342 | 0.114 | 0.174 | 0.259 | 0.149 |
| Q2B-DEEPSETS | 0.246 | 0.405 | 0.226 | 0.207 | 0.275 | 0.403 | 0.107 | 0.182 | 0.256 | 0.153 |
| Q2B-AVG-1P | 0.227 | **0.468** | 0.191 | 0.16 | 0.234 | 0.275 | 0.094 | 0.162 | **0.332** | 0.125 |
| Q2B-SHAREDOFFSET | 0.196 | 0.318 | 0.187 | 0.172 | 0.228 | 0.312 | 0.098 | 0.156 | 0.169 | 0.127 |

Table 9: MRR results of QUERY2BOX vs. several variants on FB15k, FB15k-237 and NELL995.

