# OpenReview forum: "Query2box: Reasoning over Knowledge Graphs in Vector Space Using Box Embeddings"
_ICLR.cc/2020/Conference — Accept (Poster)_

### Official Review · AnonReviewer1 · 2019-10-21
**Official Blind Review #1**

**Rating:** 8

**Review:**

The paper introduces an approach to answering queries on knowledge graphs, called Query2Box. The idea is to work with the embeddings of the vertices of the knowledge graph as if they were kind of sets. In this way, from a set, called box, of entities embeddings it is possible to project them to find other boxes using the relations specified by the query (these boxes contain the embeddings of the entities linked with those of the previous box by the relation specified in the query), or to intersect them to find the common entities.

Moreover, following this similarity, the approach is extended to solve queries containing disjunctions as well. The idea is to transform a query into its disjunctive normal form, solve each conjunction on its own (allowing the process to be parallelized) and finally answer with the set of entities in the boxes given by each conjunction.

A distance measure is used to check the belonging of an entity to a box.

This extension to disjunction could in principle be applied to other existing methods. In fact, the presented system is compared with GQE appropriately extended to handle disjunctions.

The experiments show that Query2Box can achieve better results than GQE. Moreover, an ablation study was conducted.

While I am not an expert on the subject of reasoning on knowledge graph using embeddings, the proposal seems to me to achieve interesting results, and thus to be worthy of attention in the community. Comparing Query2Box with a state-of-art system like GQE positions the new approach as a good alternative to the state of the art.

The paper is well written, there are no problems in the use of English and in the organization of the paper.

The only error I have found is on page 5, equation 4, where v'_i is used while in the text below v_i is used, so I think the two notations should be aligned.

As a final score, I would not assign a high score because I am not experienced enough in the area to be sure about the validity of the approach, which, however, seems to be good and mature.

**Experience Assessment:**

I do not know much about this area.

**Review Assessment: Checking Correctness Of Derivations And Theory:**

N/A

**Review Assessment: Checking Correctness Of Experiments:**

I carefully checked the experiments.

**Review Assessment: Thoroughness In Paper Reading:**

I made a quick assessment of this paper.

---

> ### Author Response · Authors · 2019-11-14
> **Re: Official Blind Review #1**
>
> Thank you for your positive and detailed review and constructive feedback . We are glad that the reviewer like our approach.
> Thank you for catching the typo, which we fixed.

---

### Official Review · AnonReviewer3 · 2019-10-24
**Official Blind Review #3**

**Rating:** 6

**Review:**

The paper studied the problem of answering complex logical queries on KGs, by proposing an embedding approach that encodes the queries into hyper-rectangles. The authors show that the proposed QUERY2BOX achieves the performance improvement in answering EPFO queries, as well as handling complex queries that is not observed in the training data. Experimental results show the efficacy of the proposed model. In general, I like the paper due to the nice presentation and promising approaches. However, I am not familiar with the context of KGs. I could not find anything wrong with this paper, but also do not have many intelligent questions to ask.  My only concern is the comparison experiments. The authors only presented the experiments using one baseline model GQE over two benchmark datasets. The authors may want to conduct more comparison experiments with the recent advances (e.g., Guu et al., 2015; Das et al., 2017) mentioned in the paper.

**Experience Assessment:**

I do not know much about this area.

**Review Assessment: Checking Correctness Of Derivations And Theory:**

I carefully checked the derivations and theory.

**Review Assessment: Checking Correctness Of Experiments:**

I carefully checked the experiments.

**Review Assessment: Thoroughness In Paper Reading:**

I read the paper at least twice and used my best judgement in assessing the paper.

---

> ### Author Response · Authors · 2019-11-14
> **Re: Official Blind Review #3**
>
> We thank the reviewer for the positive and detailed feedback. We are glad that the reviewer likes our approach. Below, we address the reviewer’s concern on experimental comparisons.
>
> The GQE that we primarily compared with is published in NeurIPS 2018 and is the most recent and state-of-the-art approach in handling complex multi-hop logical queries. GQE also generalizes Guu et al. 2015 (which can only handle path queries) and includes it as a special case. Das et al. 2017 built on top of Guu et al. 2015 and used LSTM to model path queries, which could potentially be integrated into our query2box framework. We leave it for future work as it is non-trivial to extend LSTM to operate on tree-structured DAG to handle conjunctive/EPFO queries.

---

### Official Review · AnonReviewer2 · 2019-10-24
**Official Blind Review #2**

**Rating:** 6

**Review:**

This paper proposes a method to answer complex logical queries in large incomplete knowledge bases (KB). Specifically it considers the class of existential first-order logical queries (EPFO) which includes the logical and, or and existential operator.  The key contribution of this paper is to represent sets of entities via regions, more specifically as boxes or hyper-rectangles. This is well motivated because such logical queries often involves working over sets of entities at once and involves applying set based operators. Previous work which represented queries as a point in vector space are not well suited for these queries.
Instead, this paper models sets of entities as boxes or axis-aligned hyper-rectangles which is parameterized by two vectors \in \mathbb{R}^{n] denoting the center and the offset respectively. Boxes can also be understood to represent all the points in it (measure by element-wise comparison with the min and max coordinate). Handling the queries require projection and intersection operation. They are defined by simple addition operation (which guarantees the boxes grow in size, due to positive offset values) and a shrinkage function to denote intersection that guarantees the output area is smaller and is inside the set of boxes.
For handling disjunctive queries, they make the clever trick of converting queries to DNF form so that the union operation is at the end of the computation graph which effectively reduces to taking the union of sets at the end.
Experiments are run on standard datasets (FB15k and FB15k-237) — however, they generate their own query patterns. Specifically, they train on 5 patterns involving projection and intersection operation and test on 4 unseen ones. For baselines, they only compare to previous work of Hamilton et al., 2018 that maps queries to vectors.

Strengths:
1. Most knowledge base comprehension benchmark tests on link prediction problems which are queries of kind (e1, r, ?). However, semantic parsers of natural language produce queries that are much richer in shape. This paper (and Hamiltion et al., 2018 before) considers answering complex logical queries (although the shape of query is pre-defined and not arbitrarily complex).
2. Modeling logical queries into regions in vector space is an interesting idea, and it would be nice to see followup work in this direction.
3. The paper is nicely written and ablation experiments were helpful.
4. Compared to the baseline they used, the paper does a better job of modeling complex logical queries.

Weaknesses / Questions:
1. I understand that the papers have considered various pre-determined shapes of queries, but the simple 1p query is similar to the usual benchmarks, I don’t understand why results for 1p were not compared with existing benchmark results. Without that comparison, I don’t have a good sense if region-based method actually are effective for “1p” kind of queries.
2. Even though the model was tested on two variants of freebase datasets, FB15k is well-known to have a lot of issues (Toutanova and Chen, 2015). Why weren't other standard datasets such as Nell-995, WN18RR and so many other biomedical KBs not considered for experiments, especially because the query generation process is very simple and it's easy to run experiments
3. It was not clear to me how the intersection operator would give zero offset for a set of non-overlapping boxes as input. Is the zero value coming from the deep-set model?, If so, how do you ensure that? Minor: Please include the deep-set network here instead of in Sec 4.3. I was confused about what the deepest model is until this point.
4. Regarding the results, is there any particular reason the MRR metric was pushed to the appendix and only results of Hits@3 was shown in the main section of the paper. I believe MRR is a better metric for your case because you are modeling sets of entities as answers and hence a ranking metric that ranks all entities is better, Hits@k is 1 if any of the answers in the set is present in top-k and hence quite a loose metric.
5. Why is the result of 3i is better than 2i. I am not sure why the model would do a better job in handling 3 intersections better than it does 2 intersections.
6. How many answers are there on an average for each question. This will better help me understand how hard the dataset actually is.
7. It is nice to see, that the model prefers boxes of different width. Do you have a sense of which type of entities (or relations) have higher offsets. This analysis would be nice to have for readers in the appendix section

**Experience Assessment:**

I have read many papers in this area.

**Review Assessment: Checking Correctness Of Derivations And Theory:**

I assessed the sensibility of the derivations and theory.

**Review Assessment: Checking Correctness Of Experiments:**

I carefully checked the experiments.

**Review Assessment: Thoroughness In Paper Reading:**

I read the paper thoroughly.

---

> ### Author Response · Authors · 2019-11-14
> **Re: Official Blind Review #2 (2/2)**
>
> RE: Please include the deepset network here instead of in Sec 4.3.
>
> Thanks for the suggestion. We included our deepsets architecture in Section 3.2 of our paper as suggested.
>
> RE: Why MRR is pushed to Appendix?
>
> This was done purely for space saving reasons. We showed H@K primarily due to the lack of space, and we observe a similar trend for MRR.
>
> RE: Why is the performance on 3i queries better than 2i queries?
>
> We agree it is counterintuitive. One possible reason why 3i queries are easier to answer is that they require modeling less number of answer entities compared to 2i queries (see the statistics below).
>
> RE: Number of answers for each query structure.
>
> The following summarizes the average number of answers in test queries grouped by different query structures. We see that complex logical queries indeed require modeling a huge number of answers than the simple 1p query does, which is exactly what motivates us to use box embeddings to perform reasoning. We shall include these statistics in the final version of the paper.
>
> FB15k
> Simple query 1p: 10.8,
> Complex queries: 2p: 255.6 / 3p: 250.0 / 2i: 90.3 / 3i: 64.1 / ip:  593.8 / pi: 190.1 / 2u: 27.8 / up: 227.0
>
> FB15k-237
> Simple query 1p: 13.3,
> Complex queries: 2p: 131.4 / 3p: 215.3 / 2i: 69.0 / 3i: 48.9 / ip:  593.8 / pi: 257.7 / 2u: 35.6 / up: 127.7
>
> NELL:
> Simple query 1p: 8.5,
> Complex queries 2p: 56.6 / 3p: 65.3 / 2i: 30.3 / 3i: 15.9 / ip: 310.0 / pi: 144.9 / 2u: 14.4 / up: 62.5
>
> We observe 2i / 3i queries have more answers than 1p queries despite the fact that intersection is taken. This is because during the query generation process, we ensured non-empty intersection in 2i / 3i queries, which biases the used relations to be one-to-many relations.
>
> RE: What type of relations have higher offset?
>
> Thanks for the insightful comments. Such an analysis is indeed helpful. Below, we demonstrate on FB15k that (1) box offset significantly varies across different relations, (2) one-to-many relations tend to have larger offset embeddings. We will include our insight in the final version of our paper.
>
> For each relation, we consider (1) the L1 norm of box offset, (2) the average number of answers of 1p queries using the relation.
>
> Top 10 relations with the smallest box size
> Relation: /architecture/structure/owner./architecture/ownership/owner
> #Avg_answers: 1.0, Box size: 2.3
> Relation: /base/petbreeds/dog_breed_group/dog_breeds
> #Avg_answers: 2.0, Box size: 4.0
> Relation: /education/educational_institution/campuses
> #Avg_answers: 1.0, Box size: 4.3
> Relation: /education/educational_institution_campus/educational_institution
> #Avg_answers: 1.0, Box size: 4.6
> Relation: /base/collectives/collective_member/member_of./base/collectives/collective_membership/collective
> #Avg_answers: 1.0, Box size: 5.1
> Relation: /base/collectives/collective/members./base/collectives/collective_membership/member
> #Avg_answers: 1.0, Box size: 5.1
> Relation: /people/appointee/position./people/appointment/appointed_by
> #Avg_answers: 1.0, Box size: 5.2
> Relation: /base/fashionmodels/hair_color/fashion_models_with_this_hair_color
> #Avg_answers: 2.0, Box size: 5.2
> Relation: /fictional_universe/fictional_character/parents
> #Avg_answers: 1.0, Box size: 5.5
> Relation: /american_football/football_player/receiving./american_football/player_receiving_statistics/team
> #Avg_answers: 2.0, Box size: 6.7
>
> Top 10 relations with largest box size
> Relation: /common/annotation_category/annotations./common/webpage/topic
> #Avg_answers: 3616.0, Box size: 147.0
> Relation: /user/tsegaran/random/taxonomy_subject/entry./user/tsegaran/random/taxonomy_entry/taxonomy
> #Avg_answers: 1.0, Box size: 137.2
> Relation: /common/topic/webpage./common/webpage/category
> #Avg_answers: 1.3, Box size: 125.6
> Relation: /base/aareas/schema/administrative_area/administrative_area_type
> #Avg_answers: 1.0, Box size: 123.6
> Relation: /medicine/drug/legal_status
> #Avg_answers: 1.5, Box size: 114.9
> Relation: /people/marriage_union_type/unions_of_this_type./people/marriage/spouse
> #Avg_answers: 889.8, Box size: 114.3
> Relation: /sports/sports_position/players./soccer/football_roster_position/team
> #Avg_answers: 397.9, Box size: 113.9
> Relation: /people/marriage_union_type/unions_of_this_type./people/marriage/location_of_ceremony
> #Avg_answers: 132.0, Box size: 108.4
> Relation: /sports/sports_position/players./sports/sports_team_roster/team
> #Avg_answers: 83.1, Box size: 104.5
> Relation: /user/tsegaran/random/subject_taxonomy/entry./user/tsegaran/random/taxonomy_entry/subject
> #Avg_answers: 495.0, Box size: 104.2
>
> We observe a clear trend that the size of the box has a strong correlation with the average number of answer entities of 1p queries using the relation.

---

> ### Author Response · Authors · 2019-11-14
> **Re: Official Blind Review #2 (1/2)**
>
> We thank the reviewer for thorough and constructive comments. Based on reviewer’s valuable feedback we conducted a number of additional experiments and provided additional dataset statistics, which further validate the efficacy of our query2box framework. We believe these together further strengthen our work.
>
> Below please find responses to individual comments/questions:
>
> RE: Lack of comparison to existing link prediction methods.
>
> This is a good point and we have conducted additional experiments. Below, we report the link prediction performance (no multi-hop logical reasoning required) following the conventional metrics (taking average over the triples). We found that our query2box is comparable or slightly better than TransE on simple link prediction. Note that in the case of simple link prediction, we do not expect a huge performance gain by using box embeddings as link prediction does not involve logical reasoning nor handling a large set of answer entities.
>
>
> FB15k			H@3		MRR
> query2box		0.613		0.516
> query2box-1p	0.633		0.531
> TransE			0.611		0.522
>
> FB15k-237		H@3		MRR
> query2box		0.331		0.295
> query2box-1p	0.323		0.292
> TransE			0.318		0.289
>
> Here query2box is trained on all five query structures as shown in Fig.4 of the paper, and query2box-1p is only trained on 1p queries. Thus, even if we train query2box over diverse queries, its performance on link prediction is still comparable to TransE, which is trained solely on the link prediction task.
>
> Although TransE is certainly not state-of-the-art in link prediction, we built on top of TransE because we want to make our methods directly comparable to existing state-of-the-art methods in logical query answering (Hamilton et al. 2018, Guu et al. 2015) that are also built on top of TransE but used single point to model queries instead of boxes. However, we emphasize that our core idea of using a box to represent a query can in principle be combined with more recent KG methods such as RotatE (Sun et al. 2019) and more advanced training techniques, which are orthogonal and we left as future work.
> We shall include these additional results in the final version of the paper.
>
> RE: Evaluation on other benchmark datasets.
>
> Thank you for your suggestion. It is always good to include more evidence to test our framework. In Appendix D, we included our additional experimental results on NELL995 dataset (a KG that contains factual knowledge about entities in the world), where answering complex logical queries is of practical interest. The new results on the additional dataset demonstrate a similar trend as the two FB15k datasets. This further validates the benefit of our query2box framework in handling complex logical queries.
>
> RE: How can intersection operator model zero offset for non-overlapping boxes?
>
> This is a good question. Our current model learns the box offset in a purely data-driven manner; thus, we do not explicitly ensure that the intersection of non-overlapping boxes has zero offset but the model can definitely learn it. In our preliminary experiments, we found our rich learnable parametrization of the box intersection consistently gives better results than strict intersection of raw boxes. This is possibly because our richer learnable parameterization of the intersection operator is more expressive and also robust to noise in the knowledge graphs. Furthermore, in principle, we can explicitly train our model on intersection queries whose answers are empty. Our deepsets (with sigmoid output activation) will then learn to output (almost) zero offset on those empty queries.
>
> In our experiments, at training time, we ensure that all intersection queries have answers, so our current model is not explicitly trained to handle empty sets. Nevertheless, we found box sizes (measured by the L1 norm of box offset) of intersection queries still offer strong insight on the number of entities they represent. Specifically, on FB15k, we randomly generate 10k queries of two types: (1) intersection queries with more than 5 answers and (2) intersection queries with empty answer (note that we have never trained on the type (2) of intersection queries). We found that the average box size is 0.36 for type (1) queries, while 0.7 for type (2) queries. Furthermore, type (2) queries are much more likely to have smaller boxes than type (1) queries (91% ROC-AUC score). This indicates that the empty intersection queries have much smaller box sizes than queries with non-zero answers.
>
> We will further clarify these points in the paper.

---

> ### Public Comment · ~Ishika_Singh1 · 2020-05-03
> **More Questions**
>
> 1. I didn't a find a mention on how do you convert natural language queries to your usable form, i.e., DNF. Do you use semantic parsing, and other resolution methods to convert using an algorithm, or is it done manually?
>
> 2. How do you find your final answer after finding a set of answers? Do you perform some query-answer matching mechanism? Could you elaborate more on range search?
>
> 3. The statistics are pretty amazing! Could you show more actual examples on the answered queries under the 9 categories that you define?
>
> 4. Could this model answer complex questions (how many, what all, as in counting all possible answers under a categorie and giving all possible outputs) as defined in Leveraging Domain Context for Question Answering Over Knowledge Graph (Tong et al., 2019) (https://link.springer.com/article/10.1007/s41019-019-00109-w)? It intuitively seems to me that it does, but I wanted see some actual examples on this?

---

### Decision · Program_Chairs · 2019-12-19

**Decision:**

Accept (Poster)

**Comment:**

This paper proposes a new method to answering queries using incomplete knowledge bases. The approach relies on learning embeddings of the vertices of the knowledge graph. The reviewers unanimously found that the method was well motivated and found the method convincingly outperforms previous work.